# Is Neoadjuvant Treatment Justified in Clinical T1 Pancreatic Ductal Adenocarcinoma?

**DOI:** 10.3390/jcm10040873

**Published:** 2021-02-20

**Authors:** Hyung Sun Kim, Kenji Nakagawa, Takahiro Akahori, Kota Nakamura, Tadataka Takagi, Masayuki Sho, Joon Seong Park, Dong Sup Yoon

**Affiliations:** 1Pancreatobiliary Cancer Clinic, Department of Surgery, Gangnam Severance Hospital, Yonsei University, Seoul 06273, Korea; MILK8508@yuhs.ac (H.S.K.); YDS6110@yuhs.ac (D.S.Y.); 2Department of Surgery, Nara Medical University, Nara 634-8521, Japan; k-nakagawa@naramed-u.ac.jp (K.N.); akahorin@naramed-u.ac.jp (T.A.); knaka@naramed-u.ac.jp (K.N.); t.takagi@naramed-u.ac.jp (T.T.); m-sho@naramed-u.ac.jp (M.S.)

**Keywords:** neoadjuvant therapy, pancreatic neoplasms, gemcitabine

## Abstract

*Introduction*: Studies on neoadjuvant treatment have been actively conducted in patients with resectable pancreatic cancer. However, neoadjuvant treatment effectiveness, especially in clinical T1 stage patients, still needs to be determined. We comparatively evaluated the oncologic benefit of preoperative neoadjuvant treatment in clinical T1 stage pancreatic cancer. *Methods:* Data from two centers were included in the comparative analysis, with overall and recurrence-free survival as primary outcomes, between January 2010 and December 2017. *Results:* In total, 45 patients were retrospectively reviewed in this study. Two patients in the neoadjuvant group were excluded because of distant metastasis during neoadjuvant treatment. Finally, 43 patients underwent a pancreatectomy for clinical T1 pancreatic cancer, of whom, 35 and 8 patients underwent upfront surgery and neoadjuvant treatment, respectively. Overall survival was similar in the two study groups (5-year overall survival rate: neoadjuvant group, 75%; upfront surgery group, 43.9%, *p* = 0.066). *Conclusions*: In our study on patients with clinical T1 stage pancreatic cancer, no significant differences were reported in the oncological outcome in the neoadjuvant therapy group. Large-scale prospective studies are needed to determine the survival benefits of neoadjuvant treatment for early-stage pancreatic cancer.

## 1. Introduction

Pancreatic ductal adenocarcinoma (PDAC) is a fatal disease. Despite advances that have facilitated early screening and diagnosis, the overall 5-year survival rate of pancreatic cancer has remained unchanged [1,2,3,4,5,6]. In resectable pancreatic cancer, the overall survival (OS) rate improved with the use of the FOLFIRINOX regimen (oxaliplatin, irinotecan, 5-fluorouracil, and l-leucovorin) as adjuvant treatment in a recent study [7]. However, even after surgery, the early recurrence rate is high, and it is difficult to ensure an R0 resection rate of 100%. Therefore, some institutions have begun to include neoadjuvant treatment for resectable pancreatic cancer on the basis of the demonstrated effect of neoadjuvant treatment in borderline resectable pancreatic cancer [8]. A database analysis of 8960 patients with pancreatic cancer found that approximately 5% of the patients had Stage IA (T1N0) disease, with an associated 5-year survival rate of 50% [9].

It is unknown whether neoadjuvant treatment is effective in patients with clinical T1 disease, and no study has investigated the efficacy of neoadjuvant therapy in early clinical T1 stage tumors. Therefore, this study was undertaken with an aim to comparatively evaluate the effect of neoadjuvant treatment with that of upfront surgery in clinical T1 PDAC.

## 2. Materials and Methods

### 2.1. Patients

We retrospectively included 45 consecutive patients who were diagnosed for clinical T1 PDAC at Gangnam Severance Hospital (*n* = 18) and Nara Medical University Hospital (*n* = 27) between January 2010 and December 2017. All patients in this study had resectable PDAC according to NCCN. Borderline PDAC cases were not included. During the study period, 35 of the patients underwent upfront surgery and 10 received neoadjuvant treatment. However, two patients in the neoadjuvant treatment group did not undergo surgery due to the detection of distant metastasis after the completion of neoadjuvant treatment. Therefore, 43 patients with clinical T1 stage PDAC underwent pancreatectomy (Figure 1). The neoadjuvant therapy regimen comprised gemcitabine and concomitant radiation up to a cumulative dose of 54 Gy. One institution involved in this study has been performing neoadjuvant chemoradiotherapy in resectable pancreatic cancer patients since 2010. In accordance with the institutional policy, neoadjuvant chemoradiotherapy was performed in all patients after 2010.

Adjuvant chemotherapy was undertaken in accordance with the institutional policies at each of the study centers. Mostly, the chemotherapy regimens were gemcitabine-based therapies (gemcitabine, 70%; gemcitabine+TS-1, 18%; and TS-1, 12%). Patients underwent a monthly follow-up during the first 3 months, and then a quarterly follow-up (every 3 months) for the next 2 years. Blood tests, including evaluation of tumor markers, were conducted every 3 months for 2 years. Abdominal computed tomography (CT) scanning was carried out every 6 months for up to 2 years. The study protocol was approved by the Institutional Review Board of the Gangnam Severance Hospital, Yonsei University of Korea (3-2020-0153) and Nara Medical University (2233). The study complied with the Declaration of Helsinki. Informed consent was obtained from all participants.

### 2.2. Tumor Assessment

The presence of a tumor was confirmed by radiological imaging techniques. Clinical T1 (tumor size <2 cm) pancreatic tumors were evaluated by radiological methods, including CT, magnetic resonance imaging (MRI), or endoscopic ultrasound (EUS). We selected patients without lymph node metastasis on preoperative imaging by CT, MRI, and positron emission tomography–computed tomography (PET-CT).

### 2.3. Statistical Analysis

All statistical analyses were conducted in SPSS, version 23.0 (SPSS Inc., Chicago, IL, USA). Categorical variables were evaluated with the chi-square or Fisher’s exact tests. We analyzed OS and recurrence-free survival (RFS) according to treatment start date. The OS and RFS were calculated from the start of chemoradiation in the neoadjuvant group, and from surgery in the upfront surgery group.

The OS and recurrence-free survival (RFS) curves were plotted using the Kaplan–Meier method, and intergroup differences in survival time were assessed with the log-rank test. RFS was defined as the interval between the date of surgery and the date of recurrence or last follow-up. OS was defined as the interval between the date of surgery and the date of death or last follow-up. A *p*-value of less than 0.05 was considered statistically significant.

## 3. Results

### 3.1. Clinical Characteristics of Patients with Pancreatic Cancer

The mean age of the patients in this study was 66.7 (SD ±10.8) years, and the study population included 24 men (55.8%) and 19 (44.2%) women. Using the classification of the American Joint Committee on Cancer (AJCC) 8th edition in the neoadjuvant group, the pathologic T stages of ypTis, ypT1, ypT2, and ypT3 were reported in 1 (12.5%), 5 (62.5%), 2 (25%), and 0 (0%) patients, respectively, and the pathologic N stages of ypN0, ypN1, and ypN2 were reported in 7 (87.5%), 1 (12.5%), and 0 (0%) patients, respectively. Subsequently, two patients from the neoadjuvant group had a higher-grade T stage tumor after resection (*n* = 2/8, 25%).

In the upfront surgery group, the pathologic T stages of Tis, T1, T2, and T3 were reported in 3 (8.3%), 26 (74.3%), 4 (11.4%), and 2 (5.7%) patients, respectively, and the pathologic N stages of N0, N1, and N2 were reported in 24 (68.6%), 8 (22.9%), and 3 (8.6%) patients, respectively. In the upfront surgery group, six patients underwent T-stage upstaging after resection (*n* = 6/35, 17%) (Table 1).

In the neoadjuvant group, the R0 resection rate was 100%. However, only 33 patients (94.3%) in the upfront surgery group showed R0 resection (*p* = 0.489). At the time of diagnosis, the levels of cancer antigen 19-19 (CA19-9) in the upfront surgery and the neoadjuvant groups ranged from 62.2 [0.8–612] to 93.7 [2–441] U/mL, respectively.

### 3.2. RFS and OS in Patients with Pancreatic Cancer

In the 43 patients who were included in the data analysis, the 1- and 3-year RFS rates were 73.4% and 55.9%, respectively. In these patients, the 1- and 3-year OS rates were 95.3% and 69.5%, respectively. The median recurrence-free survival time was 57 months, and the median overall survival time was 57.7 months.

There was no significant difference in the RFS between the two study groups (*p* = 0.171). In the neoadjuvant group, the 1- and 3-year RFS rates were 87.5% and 75%, respectively. In the upfront surgery group, the 1- and 3-year RFS rates were 70.1% and 51.2%, respectively (median recurrence-free survival time; 53 months) (Figure 2A). There was no significant difference in the OS rate between the two study groups (*p* = 0.066). In the neoadjuvant and upfront surgery groups, the 3- and 5-year OS rates were 100% and 75%, and 62.9% and 43.9%, respectively (upfront surgery groups, median overall survival time; 54.5 months) (Figure 2B).

We analyzed all 45 patients (including 2 patients with distant metastasis) to evaluate the neoadjuvant treatment effectiveness. There was no significant difference in the RFS between the two study groups (*p* = 0.604). In the neoadjuvant group, the 1- and 3-year RFS rates were 80% and 60%, respectively. In the upfront surgery group, the 1- and 3-year RFS rates were 70.1% and 51.2%, respectively (median recurrence-free survival time; 53 months) (Figure 3A). There was no significant difference in the OS rate between the two study groups (*p* = 0.208). In the neoadjuvant and upfront surgery groups, the 3- and 5-year OS rates were 90% and 56.3%, and 62.9% and 43.9%, respectively (upfront surgery groups, median overall survival time; 54.5 months) (Figure 3B).

## 4. Discussion

In the last two decades, no significant improvements in patient survival have been observed in many clinical trials of adjuvant therapy, and there has been a near-complete reliance on upfront surgery as the main treatment modality for resectable pancreatic cancer [1,2,10].

In resectable pancreatic cancer, the OS rate was shown to be improved with the use of FOLFIRINOX as adjuvant treatment in a recent study [7]. Neoadjuvant treatment is the main treatment strategy in borderline resectable pancreatic cancer [8,11,12,13]. On basis of this study, several studies on the use of neoadjuvant treatment have been actively conducted in patients with resectable pancreatic cancer [11,13,14,15,16,17,18,19,20,21,22,23]. Nevertheless, the outcomes of neoadjuvant treatment in early-stage pancreatic cancer remain unclear. In this study, we attempted to ascertain whether a difference in the survival rate exists in accordance with the effect of neoadjuvant treatment in clinical T1 PDAC. The survival rates of patients with T1 pancreatic cancer (node negative) indicated good results (5-year survival, 50%) [24,25]. The results of our data analysis indicated that the 3-year OS rate was 69.5%. Therefore, the survival rate of patients with the T1 stage disease is very high with the abovementioned treatment strategy.

Surgeons tend to prioritize surgery in order to avoid inoperable situations after neoadjuvant treatment in early-stage diseases, such as clinical T1 stage cancer. In fact, in the present study, surgery was not undertaken because of distant metastasis in 20% (two patients) of participants in the neoadjuvant group. Nonetheless, our results showed no significant difference in survival analysis between the two groups.

There are several issues with regard to neoadjuvant therapy for clinical T1 stage pancreatic cancer. A recent study reported that when clinical stage I patients underwent upfront surgery, 65.5% were upstaged to Stage IIA or higher on the final pathological report [26]. In comparison, after neoadjuvant therapy, the upstaging rate was 46.7%. With regard to upstaging in the present study, in the neoadjuvant group, we found that two patients were upstaged in the final T stage after resection (*n* = 2/8, 25%). In the upfront surgery group, six patients were upstaged in the final T stage after resection (*n* = 6/35, 17%). Therefore, there was no significant difference in the upstaging rate between the two groups.

The protocols in this study facilitated the accurate measurement of tumor size by using MRI and EUS, as well as CT scanning, and we excluded patients with lymph node metastasis on the PET-CT scan. Therefore, there was no difference in the upstaging rate in our study groups, unlike in the previous study. Therefore, in the T1 stage, it is important to determine the treatment plan after appropriate subclassification using various prognosis prediction tools. In the present study, in the clinical T1 stage of pancreatic cancer, there was no significant difference in the R0 rate among the two groups, and there was no difference in the RFS and OS between the two groups of patients.

There are several limitations to this study. This was a retrospective study, and the number of patients in the neoadjuvant treatment group was too small for an accurate comparison. However, there are not many studies that have analyzed the effect of neoadjuvant treatment of resectable pancreatic cancer so far, and in particular, no studies have analyzed the results of using only neoadjuvant treatment in early-stage cancers such as clinical T1 pancreatic cancer. Therefore, we think that this study is sufficiently new and interesting, even though it was conducted with a small number of patients.

Due to the difference in the number of patients in each group, we could not conduct a matched study analysis. Moreover, in our study, the post-operative adjuvant chemotherapy regimen types were different (e.g., gemcitabine and TS-1). This difference could have potentially influenced the oncological outcomes in the two groups. The neoadjuvant treatment regimen is an important management strategy in PDAC. Recently, FOLFIRINOX has been reported to work well; therefore, if FOLFIRINOX, and not gemcitabine, was used in this study, the treatment effect would probably vary. In Korea and Japan, only gemcitabine was available as a regimen for neoadjuvant treatment in the 2010s because of the national insurance policy. Since this study was a retrospective study from 2010 to 2017, this situation was reflected.

In our study, overall survival showed a trend toward a better survival in the neoadjuvant group, even if it was not statistically significant, and the non-significance may be explained by the small sample size and could be confirmed by larger prospective trials.

## Figures and Tables

**Figure 1 jcm-10-00873-f001:**
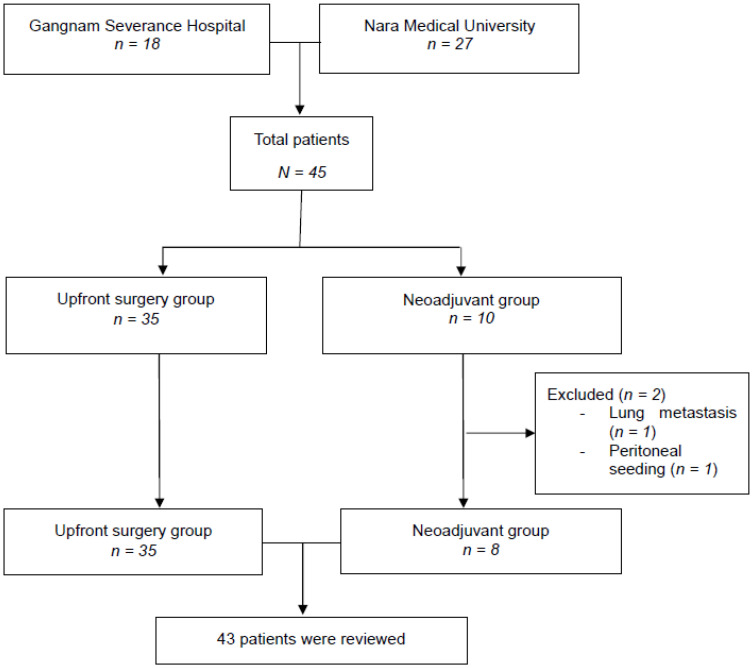
Flow diagram indicating patient disposition in the study.

**Figure 2 jcm-10-00873-f002:**
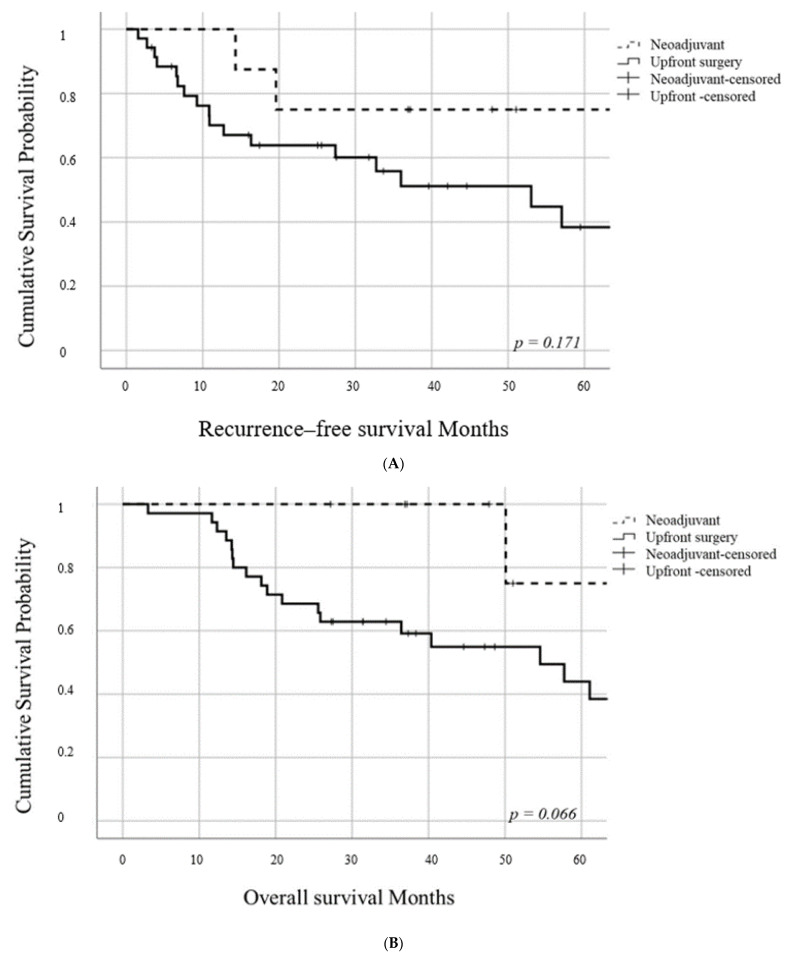
(**A**) Kaplan–Meier survival curve for recurrence-free survival. (**B**) Kaplan–Meier survival curve for overall survival.

**Figure 3 jcm-10-00873-f003:**
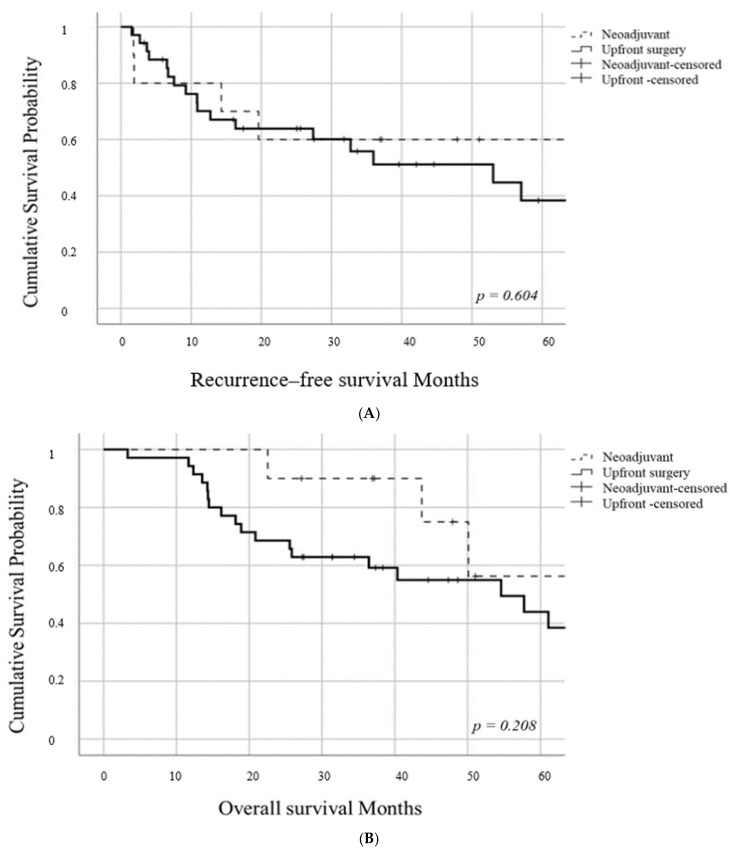
(**A**) Kaplan–Meier survival curve for recurrence-free survival in all 45 patients (including 2 patients with distant metastasis). (**B**) Kaplan–Meier survival curve for overall survival in all 45 patients (including 2 patients with distant metastasis).

**Table 1 jcm-10-00873-t001:** Clinical characteristics of patients with pancreatic cancer.

		Neoadjuvant Group (*n* = 8)	Upfront Surgery Group (*n* = 35)	*p*-Value
Age		66.1 ± 6.9	66.8 ± 11.5	0.860
Sex				0.045
	Men	7 (87.5%)	17 (48.6%)	
	Women	1 (12.5%)	18 (51.4%)	
(yp)T stage				0.671
	(yp)Tis	1 (12.5%)	3 (8.6%)	
	(yp)T1	5 (62.5%)	26 (74.3%)	
	(yp)T2	2 (25%)	4 (11.4%)	
	(yp)T3	0 (0%)	2 (5.7%)	
(yp)N stage				0.511
	(yp)N0	7 (87.5%)	24 (68.6%)	
	(yp)N1	1 (12.5%)	8 (22.9%)	
	(yp)N2	0 (0%)	3 (8.6%)	
(yp)Stage				0.897
	(yp)CIS	1 (12.5%)	3 (8.6%)	
	(yp)IA	5 (62.5%)	19 (54.3%)	
	(yp)IB	1 (12.5%)	3 (8.6%)	
	(yp)IIB	1 (12.5%)	8 (22.9%)	
	(yp)III	0 (0%)	2 (5.7%)	
PNI				0.392
	Positive	5 (62.5%)	27 (77.1%)	
	Negative	3 (37.5%)	8 (22.9%)	
LVI				0.985
	Positive	5 (62.5%)	22 (62.9%)	
	Negative	3 (37.5%)	13 (37.1%)	
Diff				0.688
	Well diff	3 (37.5%)	9 (25.7%)	
	Moderate diff	5 (62.5%)	21 (60.0%)	
	Poorly diff	0 (0%)	3 (8.6%)	
	Unknown	0 (0%)	2 (5.7%)	
Tumor size	cm	1.4 ± 0.8	1.9 ± 1.7	0.445
R0/R1				0.489
	R0	8 (100%)	33 (94.3%)	
	R1	0 (0%)	2 (5.7%)	
CA19-9	U/mL	93.7 [2–441]	62.2 [0.8–612]	0.586

PNI: perineural invasion, LVI: lymphovascular invasion, Diff: differentiation.

## Data Availability

Data sharing not applicable.

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
