# Peer review of "Is Neoadjuvant Treatment Justified in Clinical T1 Pancreatic Ductal Adenocarcinoma?"

_jcm, 2021, doi:10.3390/jcm10040873_

Round 1

Reviewer 1 Report

The Author conducted a retrospective analysis on patients with cT1 PDAC undergoing either neoadjuvamt treatment followed by surgery or upfront surgery.

The study is very interesting, however I have some considerations:

  • first of all, considering the most updated guidelines regarding the multimodal approach for PDAC, I would like to know in which situation did patients with a cT1 cancer received neoadjuvant radiochemotherapy?
    - as the Author underlined, 2/10 patients of the neoadjuvant treatment group did not undergo surgery for metastasis occurrence. I think that a 20% rate of cancer progression to metastatic disease after neoadjuvant treatment is too high to legitimate this therapeutic strategy. Moreover, these two patients should be included in the neoadjuvant group OS estimation to obtain a more precise comparison of the effectiveness of the two treatments.
  • to assess the usefulness of the neoadjuvant regimen, a lower pathological TNM staging in the radiochemotherapy group was expected but, unfortunately, the results did not show any difference.
    The results obtained by the Author do not seem to add new information to the current literature. This is due to different factors:
    -        No statistical difference was obtained when comparing the two groups in terms of RFS, OS, R0 after surgical management;
    -        A study group composed by 8 patients cannot be considered sufficiently large to start a statistical analysis;
    -        The bias represented by the different adjuvant regimen used for different patients deeply affect the results of RFS and OS.

Author Response

The Author conducted a retrospective analysis on patients with cT1 PDAC undergoing either neoadjuvamt treatment followed by surgery or upfront surgery.

The study is very interesting, however I have some considerations:

Point 1: first of all, considering the most updated guidelines regarding the multimodal approach for PDAC, I would like to know in which situation did patients with a cT1 cancer received neoadjuvant radiochemotherapy?

Response 1:

One institution involved in this study has been performing neoadjuvant chemoradiotherapy in resectable pancreatic cancer since 2010. In accordance with institutional policy, neoadjuvant chemoradiotherapy was performed in all patients after 2010.

We inserted these sentences in manuscript. ( Materials and Methods. Page 2)

Point 2 :- as the Author underlined, 2/10 patients of the neoadjuvant treatment group did not undergo surgery for metastasis occurrence. I think that a 20% rate of cancer progression to metastatic disease after neoadjuvant treatment is too high to legitimate this therapeutic strategy. Moreover, these two patients should be included in the neoadjuvant group OS estimation to obtain a more precise comparison of the effectiveness of the two treatments.

Response 2:

In Figures 3A and 3B of the results, we showed the recurrence free survival and overall survival curves analyzed including two patients with distant metastasis.

In Results section, (Page 6-7)

We analyzed all 45 patients (including 2 patient with distant metastasis) to evaluate the neoadjuvant treatment effectiveness. There was no significant difference in the RFS between the two study groups.(p=0.604) In the neoadjuvant group, the 1- and 3-year RFS rates were 80% and 60%, respectively. In the upfront surgery group, the 1- and 3-year RFS rates were 70.1% and 51.2%, respectively. (median recurrence free survival time ; 53 months) ( Figure 3A) There was no significant difference in the OS rate between the two study groups.(p=0.208) In the neoadjuvant and upfront surgery groups, the 3- and 5-year OS rates were 90% and 56.3% and 62.9% and 43.9%, respectively. (Upfront surgery groups, median overall survival time ; 54.5 months ) (Figure 3B)

Point 3: to assess the usefulness of the neoadjuvant regimen, a lower pathological TNM staging in the radiochemotherapy group was expected but, unfortunately, the results did not show any difference.

The results obtained by the Author do not seem to add new information to the current literature. This is due to different factors:

-       No statistical difference was obtained when comparing the two groups in terms of RFS, OS, R0 after surgical management;

-       A study group composed by 8 patients cannot be considered sufficiently large to start a statistical analysis;

-       The bias represented by the different adjuvant regimen used for different patients deeply affect the results of RFS and OS.

Response 3:

You are right. However, there are not many studies that have analyzed the effect of neoadjuvant treatment of resectable pancreatic cancer so far, and in particular, no studies have analyzed only neoadjuvant treatment in early stage cancers such as clinical T1 pancreatic cancer.

Therefore, we think that this study will be sufficiently new and interesting, even though it was conducted with a small number patients.

We inserted these sentences in manuscript. ( Discussion section, Page 8)

Reviewer 2 Report

The authors present a retrospective study with a small series of patients from two centers treated with upfront surgery or neoadjuvant Gemcitabine + radiotherapy for T1N0 resectable pancreatic PDAC.

The commonly accepted classification to assess resectability of PDAC is based on international consensus on NCCN classification. The authors talk about resectable PDAC in the introduction and background and then move to describe a study on T1N0 PDAC. Did these patients meet the criteria to be included in the resectable PDAC according to NCCN?

In the introduction the authors present the evidence of use of FOLFIRINOX but do not present any known results of the use of Gemcitabine, which is the protocol used in the present study. Can they incorportate current knowledge on the use of gemcitabine as neoadjuvant protocol? For example: https://doi.org/10.3389/fonc.2020.01112

The authors excluded the two patients from the neoadjuvant group who progressed during treatment.  Two out of 10 patients is a high proportion in such a small case series. Did the author perform an intention to treat analysis to assess whether including the excluded patients would affect survival results?

The authors did not report median follow up, median OS and median RFS which would be more appropriate to 1- and 3-years OS and RFS.

Interestingly there was no difference in the R0 rate among the two groups (p=0.48), whereas usually the benefit of neoadjuvant therapy in borderline resectable PDAC is associated with the highest rate of R0 post neoadjuvant. 

Can the authors clarify how they conducted the multivariate analysis on ten variables on such a small number of patients. How many events were in the neoadjuvant group? They should consider not including the multivariate analysis result in the  paper. it is statistically stretched and it does not add any novelty to the paper as it is well known from large series retrospective, prospective studies and metanalysis that R1 age and perineural invasion are independent prognostic factors.

The authors should not conclude that "In the present study, in the clinical T1 stage of pancreatic cancer, the results showed better oncological outcomes in the neoadjuvant therapy group (R0 resection rate = 100%)" as this is not accurate. Firstly, oncological results are not evaluated on the basis of the R0 rate but on survival rates. Secondly, in the present study there was no statistically significant difference in the R0 rate nor in OS and RFS.

Author Response

The authors present a retrospective study with a small series of patients from two centers treated with upfront surgery or neoadjuvant Gemcitabine + radiotherapy for T1N0 resectable pancreatic PDAC.

Point 1: The commonly accepted classification to assess resectability of PDAC is based on international consensus on NCCN classification. The authors talk about resectable PDAC in the introduction and background and then move to describe a study on T1N0 PDAC. Did these patients meet the criteria to be included in the resectable PDAC according to NCCN?

Response 1 :

Yes. All patients in this study were resectable PDAC according to NCCN. Borderline PDAC was not included.

We inserted these sentences in manuscript. ( Materials and Methods section, Page 2)

Point 2: In the introduction the authors present the evidence of use of FOLFIRINOX but do not present any known results of the use of Gemcitabine, which is the protocol used in the present study. Can they incorportate current knowledge on the use of gemcitabine as neoadjuvant protocol? For example: https://doi.org/10.3389/fonc.2020.01112

Response 2:

In Korea and Japan, only gemcitabine was available as a regimen for neoadjuvant treatment in 2010’s due to national insurance policy. Since this study is a retrospective study from 2010 to 2017, this situation was reflected.

We inserted these sentences in manuscript. ( Discussion section, Page 8)

Point 3: The authors excluded the two patients from the neoadjuvant group who progressed during treatment. Two out of 10 patients is a high proportion in such a small case series. Did the author perform an intention to treat analysis to assess whether including the excluded patients would affect survival results?

Response 3:

In Figures 3A and 3B of the results, we showed the recurrence free survival and overall survival curves analyzed including two patients with distant metastasis.

In Results section, (Page 6-7)

We analyzed all 45 patients (including 2 patient who were diagnosed during neoad-juvant treatment) to evaluate the neoadjuvant treatment effectiveness. There was no signif-icant difference in the RFS between the two study groups.(p=0.604) In the neoadjuvant group, the 1- and 3-year RFS rates were 80% and 60%, respectively. In the upfront surgery group, the 1- and 3-year RFS rates were 70.1% and 51.2%, respectively. (median recurrence free survival time ; 53 months) ( Figure 3A) There was no significant difference in the OS rate between the two study groups.(p=0.208) In the neoadjuvant and upfront surgery groups, the 3- and 5-year OS rates were 90% and 56.3% and 62.9% and 43.9%, respectively. (Upfront surgery groups, median overall survival time ; 54.5 months ) (Figure 3B)

Point 4: The authors did not report median follow up, median OS and median RFS which would be more appropriate to 1- and 3-years OS and RFS.

Response 4:

The median recurrence free survival time was 57 months and overall survival time was 57.7 months.

We inserted this sentence in manuscript. (Results section, Page 4)

Point 5: Interestingly there was no difference in the R0 rate among the two groups (p=0.48), whereas usually the benefit of neoadjuvant therapy in borderline resectable PDAC is associated with the highest rate of R0 post neoadjuvant.

Response 5:

Yes. This is because the patients in this study were clinical T1 early stage cancer patients and resectable PDAC patients according to NCCN guideline, which were different from borderline resectable PDAC patients. Although not statistically significant in our study, the results showed a better R0 resection rate in the neoadjuvant therapy group.

Point 6: Can the authors clarify how they conducted the multivariate analysis on ten variables on such a small number of patients. How many events were in the neoadjuvant group? They should consider not including the multivariate analysis result in the paper. it is statistically stretched and it does not add any novelty to the paper as it is well known from large series retrospective, prospective studies and metanalysis that R1 age and perineural invasion are independent prognostic factors.

Response 6: You are right. We deleted multivariate analysis.

We revised manuscript. ( Results section, Page 4 & 5)

RFS and OS in patients with pancreatic cancer

In the 43 patients who were included in the data analysis, the 1- and 3-year RFS rates were 73.4% and 55.9%, respectively, on univariate and multivariate analyses, without any independent factors (Table 2). In these patients, the 1- and 3-year OS rates were 95.3% and 69.5%, respectively. Furthermore, with regard to the OS, multivariate analyses showed that age >65, perineural invasion, and R1 resection were independent factors predictive of poor OS (Table 3).

->

RFS and OS in patients with pancreatic cancer

In the 43 patients who were included in the data analysis, the 1- and 3-year RFS rates were 73.4% and 55.9%, respectively, on Cox regression hazard model, ypT stage was in-dependent factor. (Table 2). In these patients, the 1- and 3-year OS rates were 95.3% and 69.5%, respectively. The median recurrence free survival time was 57 months and overall survival time was 57.7 months. Furthermore, with regard to the OS, the results showed that ypT stage, perineural invasion, and R1 resection were independent factors of poor OS (Table 3).

Table 2.Univariate and multivariate analyses of the relationship between recurrence-free survival and clinicopatho-logical variables on analysis in the Cox regression hazard model.

-> Table 2. The relationship between recurrence-free survival and clinicopathological variables on analysis in the Cox regression hazard model.

Table 3. Univariate and multivariate analyses of the relationship between overall survival and clinicopathological variables on analysis in the Cox regression hazard model.

-> Table 3. The relationship between overall survival and clinicopathological variables on analysis in the Cox regression hazard model.

Point 7: The authors should not conclude that "In the present study, in the clinical T1 stage of pancreatic cancer, the results showed better oncological outcomes in the neoadjuvant therapy group (R0 resection rate = 100%)" as this is not accurate. Firstly, oncological results are not evaluated on the basis of the R0 rate but on survival rates. Secondly, in the present study there was no statistically significant difference in the R0 rate nor in OS and RFS.

Response 7: You are right. We revised these sentences in manuscript. (Discussion section, Page 8)

In the present study, in the clinical T1 stage of pancreatic cancer, the results showed better oncological outcomes in the neoadjuvant therapy group (R0 resection rate = 100%).

-> In the present study, in the clinical T1 stage of pancreatic cancer, the results showed better R0 resection rate in the neoadjuvant therapy group.

In our study on patients with clinical T1 stage pancreatic cancer, better oncological outcomes were reported in the neoadjuvant therapy group.

-> In our study on patients with clinical T1 stage pancreatic cancer, no significant differences were reported in the oncological outcome in the neoadjuvant therapy group.

Round 2

Reviewer 1 Report

The Authors did what was requested.

Author Response

The Authors did what was requested.

Response :

Thank you very much.

Reviewer 2 Report

Thank you for providing a new version of the manuscript for review.

I feel that some of the points of my previous review were not fully addressed. 1. Ref 190-191: "In the present study, in the clinical T1 stage of pancreatic cancer, the results showed better R0 resection rate in the neoadjuvant therapy group." The R0 versus R1 p value was 0.489 (not significant) So I believe this should be changed to  "there was no significant difference in R0 rate among the two groups"

2. I suggested to remove the multivariate analysis, and you replied you agreed but still presented the results of the cox regression analysis. As per my previous review, I believe this should be removed from the results as it is based on a small sample size and limited number of events. Furthermore, it does not add any novelty to what is commonly known in the published literature. I believe the paper would read better without the cox regression analysis.

3. I would suggest giving more emphasis to the OS Kaplan Meier where even if not statistically significant there seem to be a trend towards a better survival in the neoadjuvant group, and the non significance may be related to the small sample size and could be confirmed on larger prospective trials.

Author Response

Thank you for providing a new version of the manuscript for review.

I feel that some of the points of my previous review were not fully addressed.

Point 1: Ref 190-191: "In the present study, in the clinical T1 stage of pancreatic cancer, the results showed better R0 resection rate in the neoadjuvant therapy group." The R0 versus R1 p value was 0.489 (not significant) So I believe this should be changed to "there was no significant difference in R0 rate among the two groups"

Response 1:

We revised this sentence in manuscript. ( Discussion section, Page 7)

In the present study, in the clinical T1 stage of pancreatic cancer, the results showed better R0 resection rate in the neoadjuvant therapy group

  • In the present study, in the clinical T1 stage of pancreatic cancer, there was no significant difference in R0 rate among the two groups.

Point 2: I suggested to remove the multivariate analysis, and you replied you agreed but still presented the results of the cox regression analysis. As per my previous review, I believe this should be removed from the results as it is based on a small sample size and limited number of events. Furthermore, it does not add any novelty to what is commonly known in the published literature. I believe the paper would read better without the cox regression analysis.

Response 2: You are right.

We deleted cox regression analysis in manuscript. ( Results section, Table 2&3 )

Point 3: I would suggest giving more emphasis to the OS Kaplan Meier where even if not statistically significant there seem to be a trend towards a better survival in the neoadjuvant group, and the non significance may be related to the small sample size and could be confirmed on larger prospective trials.

Response 3:

We revised this sentence in manuscript. ( Discussion section, Page 7 )

In our study on patients with clinical T1 stage pancreatic cancer, no significant dif-ferences were reported in the oncological outcome in the neoadjuvant therapy group. Large-scale prospective studies are needed to determine the survival benefits of neoadju-vant treatment for early-stage pancreatic cancer.

  • OS Kaplan Meier where even if not statistically significant there seem to be a trend towards a better survival in the neoadjuvant group, and the non significance may be related to the small sample size and could be confirmed on larger prospective trials.
